# Impact of Glucocorticoids on Cardiovascular System—The Yin Yang Effect

**DOI:** 10.3390/jpm12111829

**Published:** 2022-11-03

**Authors:** Chase Kelley, Jonathan Vander Molen, Jennifer Choi, Sahar Bhai, Katelyn Martin, Cole Cochran, Prasanth Puthanveetil

**Affiliations:** 1Chicago College of Osteopathic Medicine, Midwestern University, Chicago, IL 60515, USA; 2Rm-322-I, Science Hall, Department of Pharmacology, College of Graduate Studies, Midwestern University, Chicago, IL 60515, USA

**Keywords:** glucocorticoids, systemic metabolism, cardiovascular health

## Abstract

Glucocorticoids are not only endogenous hormones but are also administered exogenously as an anti-inflammatory and immunosuppressant for their long-term beneficial and lifesaving effects. Because of their potent anti-inflammatory property and ability to curb the cytokines, they are administered as lifesaving steroids. This property is not only made use of in the cardiovascular system but also in other major organ systems and networks. There is a fine line between their use as a protective anti-inflammatory and a steroid that could cause overuse-induced complications in major organ systems including the cardiovascular system. Studies conducted in the cardiovascular system demonstrate that glucocorticoids are required for growth and development and also for offering protection against inflammatory signals. Excess or long-term glucocorticoid administration could alter cardiac metabolism and health. The endogenous dysregulated state due to excess endogenous glucocorticoid release from the adrenals as seen with Cushing’s syndrome or excess exogenous glucocorticoid administration leading to Cushing’s-like condition show a similar impact on the cardiovascular system. This review highlights the importance of maintaining a glucocorticoid balance whether it is endogenous and exogenous in regulating cardiovascular health.

## 1. Introduction

Glucocorticoids are innately known as the stress hormones which are also endogenous anti-inflammatory agents. Their presence in systemic circulation and tissues at various stages of the life cycle, from development to adulthood, have been extensively studied. GCs belong to the class of essential hormones required for our growth and systemic and metabolic homeostasis [1,2]. At the same time, when it is present in excess, it could trigger some detrimental effects on cardiovascular and systemic health [3,4,5,6,7,8,9,10,11,12,13,14,15,16,17,18]. In this review, we will go in detail to analyze the Yin-Yang effect of GCs at various life stages to understand how important it is to maintain a GC homeostasis to preserve cardiovascular health. 

## 2. Role of GC during Stages of Development

Glucocorticoids are essential during organismal and organ development as demonstrated by studies using models with either depletion of hormone, or lack of glucocorticoid receptor [2,19,20,21]. At the same time, excess hormone secretion or unregulated receptor function due to either increased expression or transcriptional activity can bring about detrimental effects [3,4,5,7,8,9,10,11,12,13,14,15,16,17,18,21]. Studies have demonstrated that women were administered synthetic glucocorticoids during mid to late gestation period as a means to increase the survival rate of the fetus due to pre-term delivery [22]. Synthetic glucocorticoids have also been proven useful for pregnant women suffering from auto-immune diseases [23,24,25,26,27,28,29,30,31], including lupus and asthma and at the same time minimizing the risks in infants with low birth weight [32,33]. There is enough evidence to suggest that clinically short-term administration of glucocorticoids is beneficial for pregnant women as stated above. In rats, administration of glucocorticoids at a dose between 100 and 200 ug/kg per day during late gestation upregulated the expression of Ca^2+^ binding proteins, calreticulin and calsequestrin [34], in the 21-day-old fetus and in adulthood [34,35]. Enhanced Ca^2+^ binding protein levels have been associated with calcium dysregulation and premature cell death [34]. Additionally, exposure of developing cardiac cells to excess GC have also been shown to cause cardiomyocyte hyperproliferation, with irregular cardiac muscle structure and potentially triggering internal apoptosis mechanisms [35]. GC administration in day 17 of gestation for rats resulted in larger hearts with increased cardiomyocyte proliferative index, with enhanced increased cardiac cell generation [36]. In a study involving a sheep model, maternal hydrocortisone (80 mg/day) administration from Day 119 resulted in increased fetal heart weight and thickened right and left ventricular walls [37]. Similarly, betamethasone administration following induction of intrauterine growth restriction (IUGR) in late-gestation sheep fetuses increased responsiveness of left ventricular Beta-adrenoceptors [37], eventually increasing susceptibility to adult cardiac dysfunction [37]. Enhanced GC exposure leads to upregulated gene expression of Ca2+ binding proteins (CaBPs), abnormal cardiac cell proliferation, and increased left ventricular wall thickening in the fetuses as demonstrated using various animal models above [34,36,37]. These studies urge us to investigate in detail the short- and long-term effects of glucocorticoid use along with their doses to understand the impact of glucocorticoid exposure during fetal development. Kim et al. demonstrated the effect of maternal dexamethasone administration on maturity of pre-term piglet’s hearts [38]. The different treatment groups included (a) term piglets delivered by C-section 2 days before farrowing date, (b) the second group was pre-term piglets delivered on the 91st day (term = 115 days) and (c) the third group was delivered on the 91st day, but the mothers were given dexamethasone 48 and 24 h before delivery similar to women delivering pre-term in a clinical scenario [38]. In total, 12 piglets (6 male and 6 female) were studied in each group. The pre-term GC administered piglets had an increased number of binucleated myocytes and increased myocyte volume [38]. How this binucleated state of myocyte with increased volume could help during postpartum state is not well defined and needs more clarity. Additionally, glucocorticoid exposed preterm hearts had a greater ability to maintain aortic flow in the face of increasing afterload compared to untreated preterm piglets’ hearts [38]. It was also observed that GC administration promotes structural maturity.

## 3. Role of GC in Heart at Young Stage or in Pediatric Population

Hailey Blain et al. reported that in a pediatric population with Cushing’s syndrome there is over risk of cardiovascular complications, increasing quadruple times in comparison to healthy children [39]. This study demonstrated a positive correlation between Cushing’s syndrome and mortality due to cardiovascular disease. There were evident features of cardiovascular remodeling starting from an early age (6 yrs) accompanied by hypertension, hyperlipidemia and deposition of connective tissue in macro vessels including aorta. In a similar manner, deficiency of this hormone also has drastic and even fatal consequences [39]. Studies involving a rat model has demonstrated that even early exposure to GC excess could contribute leads to reduced ejection fraction by 50 weeks old and with a reduced systolic function at 80 weeks old [40]. One of the reasons could be due to reduced mitotic function of cardiac cells which could then trigger compensatory hypertrophic signals contributing towards ventricular dysfunction with age [40]. Does GC excess interfere with the cardiac cell reserve during senility is yet to be determined. In a clinical study involving pediatric patients with Cushing’s syndrome [41], based on coagulation profiling it was found that patients suffer from hypercoagulable state even in the presence of excess anticoagulation factors [41]. This study brings to light the coagulation risks associated with hypercortisolemia [41]. In a rodent study, where dexamethasone was administered postnatally from day 1 to 3, the GC exposed rat hearts demonstrated increase in heart weight and heart to body weight ratio, in comparison to control group [42]. Additionally, this effect was reversed using glucocorticoid receptor inhibitor, RU486 [42]. Interestingly epigenetic regulation of heart following DEX treatment was also noted [42]. Inhibitor of methylation, 5-aza-2’-deoxycytidine (5-AZA) analog was able to decrease both heart weight and heart to body weight ratio [42]. These studies confirm the epigenetic regulation of GR function in pediatrics or young population [42]. At the same time, even the deficiency of this hormone is accompanied by detrimental health effects. Premature mortality with high risk of cardiovascular disease and cancer has been reported in patients with Addison’s disease based on clinical studies. 

## 4. Role of GC in Adult Hearts

Synthetic glucocorticoids are widely used as an anti-inflammatory for chronic use due to its ability to curb the generation of inflammatory markers [43]. This is made possible mostly due to its transcriptional control of genes involved in inflammation [43,44]. Synthetic glucocorticoids such as Dexamethasone have been shown to be effective in controlling sepsis-induced cardiac dysfunction. In a Wistar rat model of cecal ligature [45] and puncture-induced septic shock [45,46], dexamethasone was able to improve both the vascular and cardiac function improving blood pressure specifically hypotension induced by shock, normalizing systolic and diastolic functions [45]. Jointly, Dex and human activated protein C was able to reduce the peroxynitrite generation [47]. Some of the cardiovascular benefits were attributed to this peroxynitrite inhibitory effect [47]. The independent protective effects of synthetic glucocorticoids on cardiovascular benefits observed are not evaluated thoroughly. In a clinical study involving patients with cardiac amyloidosis, dexamethasone with oral melphalan and thalidomide exhibited significant improvement [48]. These patients also belonged to Class IV of New York heart failure category [48]. One of the key features of the outcome was that inclusion of dexamethasone in the treatment regimen was able to improve cardiac dysfunction in 1/5th of the participants, along with the increase in survival rate over 2 years [48]. This study also highlights the significance of using dexamethasone (synthetic GCs) as co-therapy with other agents in treating inflammatory conditions for patients suffering from cardiovascular complications.

In a study involving both male and female rats with myocardial ischemia, dexamethasone administration at a dose of 0.1 mg/kg was able to regulate the ultrastructure of the muscle [49]. Following EM-based imaging, dexamethasone treatment in the presence of ischemia demonstrated normalization of the structure integrity of the cell and the organelles, especially the mitochondrial structure [49]. This study provides validation of the cardiac mitochondrial protective role of glucocorticoids under ischemic or hypoxic conditions in the heart [49]. Along with adapting to a stress response, glucocorticoids play an important role in protecting vital organs from inflammation due to pathogen-induced infections and sterile inflammation [50]. A study also demonstrated using isolated primary ventricular cardiomyocytes that glucocorticoids through the mediation of AMPK offers protection against TNF α-induced cell death, mostly apoptosis. In patients diagnosed with cardiac sarcoidosis, patients were administered prednisolone at a dose of 60 mg on alternate days for a period of 2 months followed by tapering of dose to 10 mg administered every other day. Even though no significant changes were observed in LV volume and function, the left ventricular ejection fraction (LVEF) increased. 

As it is essential to have a normal GC signaling for proper functioning of the physiological systems, it is also very important to have a balance. Excess GCs whether endogenous or exogenous, can also be detrimental in adult cardiovascular system. A study conducted in rodents where they were exposed to chronic and variable stressors such as restraints, cold room, shaker platform or even hypoxia, and the physiological responses tracked using implanted radio telemetry transmitters. It was noted that longer exposure to these variable stress factors resulted in enhanced heart rate and blood pressure. Additionally, the stressors were able to promote collagen deposition on the vessel wall accompanied by enhanced plasma cytokine levels. Endogenously, along with GCs, 11 Beta HSD1, the enzyme involved in activating endogenous corticosteroids, present mostly in the liver also played a major role in GC mediated effects. It was also demonstrated that 11 Beta HSD1 expressing in vascular endothelium was involved in potentiating the GC effects with resultant hypertrophy and fibrosis. 

Kitterer et al. evaluated the long-term effects that glucocorticoid use would have on pericardial and epicardial fat levels using cardiovascular Magnetic Resonance (CMR). The results were based on observations from 61 individuals who had been were separated into a high dose steroid (>7.5 mg of prednisone or prednisone equivalent per day for at least 6 months before CMR) and a low dose steroid (<7.5 mg of prednisone or prednisone equivalent per day) for at least 6 months before CMR-based evaluation. A group that did not receive any steroids with matching, age, sex, and BMI served as the control group. Patients receiving high dose of GC had significantly more epicardial and pericardial fat than patients administer low dose steroid and the control group. However, significant difference was found between fat levels of low dose steroid group and the control group. “Cardiac fat distribution correlates with BMI in steroid-treated patients, but not in controls”. Based on these findings, it may be safe to assume that patients with rheumatic disorders on long-term steroid therapy suffer from increased cardiac fat deposition and it is advisable to administer a low dose of GC in these patients. Additionally, from a clinical perspective the tendency towards fat deposition in vital organs including cardiac tissue should be stringently monitored.

A UK-based cohort study using record linkage database demonstrated that, when exogenous glucocorticoid use (at a dose of >7.5 mg of prednisolone equivalent) was associated with enhanced cardiovascular risks [51]. A population-based case–control study based on nationwide databases available in Denmark demonstrated enhanced risk of venous thromboembolism with glucocorticoid use [52]. This correlation was strong in glucocorticoid use especially 90 days or less [52]. A population-based cohort study which was conducted in identifying correlation between dose dependent glucocorticoid use and cardiovascular risks in patients treated for six different immune diseases such as giant cell arteritis, polymyalgia rheumatic, inflammatory bowel disease, rheumatoid arthritis, systemic lupus erythematosus, and/or vasculitis [53]. Based on the study, even low dose of glucocorticoids, as low as <5 mg enhanced five-year cumulative risk of getting a cardiovascular disease in comparison to non-glucocorticoid users [54]. Following autoimmune disease such as multiple sclerosis, when a high dose of methylprednisolone over 1000 mg was administered for 5 days/week for either a week or three months, GCs demonstrated some cardiovascular complications in these patients [55]. An increase in systolic and diastolic pressure, decrease in flow mediated dilation of the brachial artery, left ventricular diastolic dysfunction, and endothelial dysfunction as major noted cardiovascular adverse effects always observed with chronic glucocorticoid use [53,56]. Based on the above two scenarios, it suggests that irrespective of the dose and duration of administration of GCs, another important factor we need to consider is their specificity. Glucocorticoid receptor specific ligands such as dexamethasone will fulfill the clinical requirements with minimal or negligible impact on mineralocorticoid, estrogen or other steroid receptors minimizing systemic side effects.

## 5. Recent Role of GCs in Combating Viral Infections Including COVID-19—Impact and Consequences

One of the earlier studies in mice model had demonstrated that lymphocytic choriomeningitis virus infection was able to raise endogenous corticosteroid levels as a counter-regulatory anti-viral defense [57]. This earlier work was also supported by a recent work published more than a decade later which corroborated the same finding that glucocorticoid-induced TNF alpha receptor related protein (GITR) provided anti-viral defense and potentiated the anti-viral property of helper CD8 T cells [58]. With the advent of COVID-19 infection, earlier attempts in administrating glucocorticoids for COVID-19-related co-morbidities had proven to be quite protective. A low to moderate dose of glucocorticoids have demonstrated improved symptoms and reduced hospitalization [59]. Additionally, case with exacerbated immune thrombocytopenia along with COVID-19 has demonstrated improvement with GC administration [60]. All these earlier case reports led way to larger clinical trials using GCs to curb these co-morbidities in the presence of COVID-19 infection. A single-blind, randomized controlled trial with 68 patients demonstrated that when administered synthetic glucocorticoid, methylprednisolone, patients had lower mortality and better survival rate (PMID: 32943404) [61]. Another study, which was a parallel, double-blind, placebo-controlled, randomized, Phase IIb clinical trial in hospitalized patients (approximately 393) of age ≥ 18 years with clinical symptoms of COVID-19 [62]. The subjects randomly received either intravenous administration of methyl prednisolone (0.5 mg/kg) or placebo (saline solution) twice daily for a period of five days [62]. The all-cause mortality rates at day 28 was determined. In the lower age group, no major difference was noted in all-cause mortality, but at ages above 60 years, methyl prednisolone reduced all-cause mortality at day 28 [62].

A prospective meta-analysis study [63] with collected data from 1703 patients from 12 countries from February to June 2020 based on seven different randomized trials revealed that systemic corticosteroids administration decreased 28 day all-cause mortality [63]. A controlled, open-label trial with 2104 patients administered dexamethasone and 4321 patients assigned to regular care were compared [64]. Dexamethasone administration reduced 28 days all-cause mortality among patients receiving either invasive mechanical ventilation or oxygen supplementation per se but it did not change the end results in patients who did not receive any respiratory support [64]. Regarding the impact of GCs on cardiac tissue, a retrospective study with, 72 subjects of mean age 59.6 years, with 69% male patients and rest females. Data were collected during their hospital admission for COVID-19-related symptoms and 99 days after their discharge from hospital [65]. Based on the medications administered to these patients, the ones administered dexamethasone exhibited a reduction in epicardial adipose tissue inflammation [65]. An enhanced epicardial adipose tissue inflammation could pose serious risk for patients suffering from cardiovascular disease. This trial revealed that targeting epicardial tissue inflammation following COVID-19 infection could be an ideal for protecting cardiovascular protection. Further studies are needed on these lines that could reveal specifically cardioprotective consequences of glucocorticoid administration during COVID-19 infection.

## 6. Conclusions

Our review provides succinct information regarding the relevance of endogenous glucocorticoids and their mandatory requirement during development and early stages. It also highlights the impact of excess glucocorticoids, both endogenous and exogenous on systemic health, specifically on the cardiovascular system (Figure 1). With their essential role in development and health [66], having a deeper understanding of how to maintain homeostasis of glucocorticoid levels and their effects is crucial (Figure 1). Different scenarios presented also gives a perspective regarding how important it is to maintain optimum dose, duration and select an exogenous GC which GR selective or specific to avoid undesired systemic side effects. With their recent intervention in COVID-19-related therapies specially to curb respiratory distress with resultant protective effects [64] on other major organ systems such as the cardiovascular system propel the need to understand their unexplored pharmacological effects. Overall, following ligand binding, GR undergoes nuclear translocation to bind their responsive elements in DNA to either induce or suppress their target genes bringing about either positive (under GC requirement) or negative (under GC excess conditions) impact on cardiovascular tissue (Figure 2). An in-depth understanding of cellular actions of GCs accompanied by their systemic physiological and metabolic effects will help us in handling this versatile hormone to our benefits maintaining health and homeostasis, especially in the cardiovascular system.

## Figures and Tables

**Figure 1 jpm-12-01829-f001:**
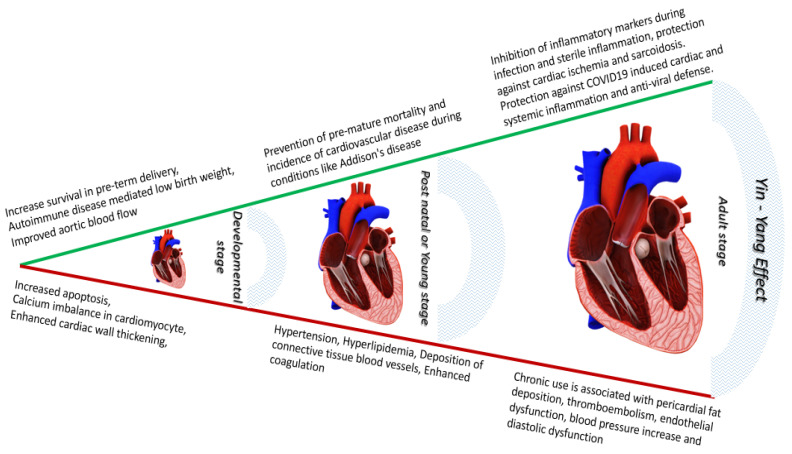
The Yin-Yang effect of glucocorticoids on cardiovascular system—glucocorticoids whether endogenous or exogenous have an inevitable role from developmental stage to post-natal and adult stages. GCs provide crucial protection to cardiovascular system against autoimmune attack on cardiovascular system during all the human life stages combined. The role of GCs against COVID-19 mediated attack on cardiovascular system is a newly identified role. In contrast, excess GC level could lead to cardiac cell apoptosis, cardiac wall thickening, along with hypertension, hyperlipidemia and endothelial dysfunction when exposed to GCs chronically debilitating cardiovascular function.

**Figure 2 jpm-12-01829-f002:**
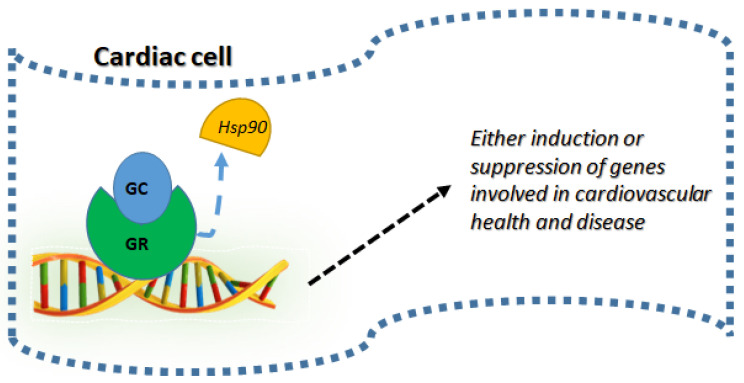
Cellular mechanism of GC on cardiovascular tissue: Following the ligand binding (GC) to GR, they detach from the chaperones such as heat shock protein 90 (Hsp90) undergoing nuclear translocation and DNA binding to either induce or suppress gene targets which are involved in cardiac health and diseases.

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
