# Peer review of "Impact of Glucocorticoids on Cardiovascular System—The Yin Yang Effect"

_jpm, 2022, doi:10.3390/jpm12111829_

Round 1

Reviewer 1 Report

The topic is actual and may add some crucial information in the field of using Glucocorticoids in patients with cardiovascular diseases.

The part (about 28%) of references is more than 10 years, so, it would be the good way to update the references list 

Author Response

Reviewer -1:

The topic is actual and may add some crucial information in the field of using Glucocorticoids in patients with cardiovascular diseases.

1 . The part (about 28%) of references is more than 10 years, so, it would be the good way to update the references list 

Response – We appreciate the reviewer for the positive comments and also for the input provided.  As per the suggestion, we have now added new references (from less than 10 years) along with the previously present ones. Altogether, we have now added 22 new references and our total references have gone up from 45 to 66.

Reviewer 2 Report

The present article is an overview of the effect of GCs on cardiovascular system and goes in detail to analyze the Yin-Yang effect of GCs at various life stages to understand how important it is to maintain a GC homeostasis to preserve cardiovascular health. The article is well written and different from the existing ones. The figure provided is clear. 

However I believed that and additional figure explaining the cellular effects of GCs on cardiomyocytes will help the readers in a better understanding of the article

Author Response

Reviewer – 2:

The present article is an overview of the effect of GCs on cardiovascular system and goes in detail to analyze the Yin-Yang effect of GCs at various life stages to understand how important it is to maintain a GC homeostasis to preserve cardiovascular health. The article is well written and different from the existing ones. The figure provided is clear. 

1 . However, I believed that an additional figure explaining the cellular effects of GCs on cardiomyocytes will help the readers in a better understanding of the article

Response – We thank the reviewer for the positive comments. In our revised version, we have now added an additional figure (figure-2) explaining the cellular actions of GC, which can help the reader in understanding how GC brings about these effects on cardiovascular system.
